# Identification of Lifestyle Behaviors Associated with Recurrence and Survival in Colorectal Cancer Patients Using Random Survival Forests

**DOI:** 10.3390/cancers13102442

**Published:** 2021-05-18

**Authors:** Moniek van Zutphen, Fränzel J. B. van Duijnhoven, Evertine Wesselink, Ruud W. M. Schrauwen, Ewout A. Kouwenhoven, Henk K. van Halteren, Johannes H. W. de Wilt, Renate M. Winkels, Dieuwertje E. Kok, Hendriek C. Boshuizen

**Affiliations:** 1Division of Human Nutrition and Health, Wageningen University & Research, 6708 WE Wageningen, The Netherlands; moniek.vanzutphen@wur.nl (M.v.Z.); franzel.vanduijnhoven@wur.nl (F.J.B.v.D.); vera.wesselink@wur.nl (E.W.); renate.winkels@wur.nl (R.M.W.); dieuwertje.kok@wur.nl (D.E.K.); 2Department of Gastroenterology and Hepatology, Bernhoven Hospital, 5406 PT Uden, The Netherlands; r.schrauwen@bernhoven.nl; 3Department of Surgery, Ziekenhuis Groep Twente, 7600 SZ Almelo, The Netherlands; E.Kouwenhoven@zgt.nl; 4Department of Internal Medicine, Admiraal de Ruyter Ziekenhuis, 4462 RA Goes, The Netherlands; hk.vanhalteren@adrz.nl; 5Department of Surgery, Radboud University Medical Centre, 6500 HB Nijmegen, The Netherlands; hans.dewilt@radboudumc.nl

**Keywords:** colorectal cancer, survival, recurrence, lifestyle, random survival forests

## Abstract

**Simple Summary:**

Current lifestyle recommendations for cancer survivors are the same as those for the general public to decrease their risk of cancer. However, it is unclear what kind of lifestyle behaviors are important for prognosis after a cancer diagnosis. In an observational study among 1180 colorectal cancer patients, we aimed to identify which lifestyle behaviors were most important regarding cancer recurrence and all-cause mortality. We simultaneously evaluated lifestyle behaviors, related to diet, physical activity, adiposity, alcohol use, and smoking. Higher intakes of sugary drinks were associated with increased recurrence risk. For all-cause mortality, fruit and vegetable, liquid fat and oil, and animal protein intake were identified as important lifestyle behaviors. Our exploratory findings identified several lifestyle behaviors related to prognosis after colorectal cancer. These findings should be confirmed in other observational studies before they can be translated into clinical practice.

**Abstract:**

Current lifestyle recommendations for cancer survivors are the same as those for the general public to decrease their risk of cancer. However, it is unclear which lifestyle behaviors are most important for prognosis. We aimed to identify which lifestyle behaviors were most important regarding colorectal cancer (CRC) recurrence and all-cause mortality with a data-driven method. The study consisted of 1180 newly diagnosed stage I–III CRC patients from a prospective cohort study. Lifestyle behaviors included in the current recommendations, as well as additional lifestyle behaviors related to diet, physical activity, adiposity, alcohol use, and smoking were assessed six months after diagnosis. These behaviors were simultaneously analyzed as potential predictors of recurrence or all-cause mortality with Random Survival Forests (RSFs). We observed 148 recurrences during 2.6-year median follow-up and 152 deaths during 4.8-year median follow-up. Higher intakes of sugary drinks were associated with increased recurrence risk. For all-cause mortality, fruit and vegetable, liquid fat and oil, and animal protein intake were identified as the most important lifestyle behaviors. These behaviors showed non-linear associations with all-cause mortality. Our exploratory RSF findings give new ideas on potential associations between certain lifestyle behaviors and CRC prognosis that still need to be confirmed in other cohorts of CRC survivors.

## 1. Introduction

The rates of cancer survival are increasing with more people living with and beyond cancer, including colorectal cancer (CRC) [1,2]. Current lifestyle recommendations for cancer survivors are the same as those for the general public to decrease their risk of cancer [3,4]. The current guidelines are to: (1) achieve and maintain a healthy body weight; (2) engage in regular physical activity; and (3) achieve a dietary pattern high in vegetables, fruits, and whole grains while limiting fast foods, red and processed meat, sugary drinks, and alcohol consumption.

Cancer patients with a healthy lifestyle after CRC diagnosis may have a better prognosis. Several meta-analyses concluded that higher levels of physical activity after CRC diagnosis were associated with lower mortality [5,6,7,8,9]. Additionally, several studies showed that body mass index (BMI) after CRC diagnosis seems associated with mortality. The risk of death was highest among patients who were underweight, while lowest risk was seen in patients with a BMI between 25 and <30 kg/m^2^ [10,11,12,13]. Although the number of studies that assessed the association between diet after CRC diagnosis and mortality is limited, it seems that healthier diets are associated with lower mortality [10]. A higher intake of fruit and vegetables [14,15,16] and wholegrains [15,17,18] were generally associated with lower mortality, although not in all studies. An unhealthy (“Western”) dietary pattern [15,19,20] or higher intake of sugary drinks [15,21] were generally associated with higher mortality. In contrast, there is currently little evidence to support the recommendation to limit red and processed meat intake after CRC diagnosis [14,15,22]. Given the limited number of studies, it remains unclear whether lifestyle after CRC diagnosis is associated with recurrence risk [10]. A limitation of these studies is that they examine the importance of single lifestyle behaviors. However, lifestyle is multidimensional, with behaviors representing dietary habits, alcohol use, physical activity, adiposity, and smoking. Considering different lifestyle behaviors simultaneously, rather than a series of separate characteristics, could provide a more comprehensive understanding of which aspects of lifestyle are most important in relation to CRC prognosis.

Efforts to quantify overall lifestyle in CRC survivors have been limited to assessing adherence to lifestyle recommendations [14,23,24]. Previous studies among CRC survivors showed that an overall healthy lifestyle after CRC diagnosis was associated with improved survival [14,24], but not among long-term survivors [23]. The results regarding CRC recurrence were inconsistent [14,24]. Our group reported that post-diagnosis lifestyle might be more important than lifestyle before diagnosis, as the summary lifestyle score before CRC diagnosis was not associated with all-cause mortality [24]. There have been no studies that have identified which post-diagnosis lifestyle behaviors are most important in relation to mortality or recurrence.

To date, researchers have used multivariable Cox proportional hazard regression models to test the hypothesis that a certain lifestyle behavior (or lifestyle score) is associated with CRC outcomes. Cox regression models can also be used to identify variables of interest. However, exploratory analysis of a dataset containing many correlated variables has statistical challenges, including correction for multiple testing and handling of multicollinearity. Random survival forests (RSF) [25] are a robust alternative for Cox regression models in the case of exploratory analyses. RSF seeks to identify a model that best explains the data, thus, before building the model there is no need to select a limited number of variables of interest or to know the relationship (i.e., linear, nonlinear) of a variable with the outcome. Furthermore, RSF can handle many variables, take complex interactions between variables into account, and does not rely on *p*-values. RSF has been successfully applied in identifying the risk factors of different diseases and disease outcomes [26,27,28,29,30,31], but has not been used to identify important lifestyle behaviors with regard to cancer prognosis.

We aimed to identify which lifestyle behaviors were most important regarding colorectal cancer (CRC) recurrence and all-cause mortality among CRC survivors with stage I-III disease with RSF. We evaluated the lifestyle behaviors that are considered in cancer prevention recommendations, as well as other lifestyle behaviors that might need to be included in future recommendations for cancer survivors.

## 2. Methods

### 2.1. Study Population

We used data from the COLON study, a prospective multicenter cohort study among CRC patients [32]. From 2010 onwards, newly diagnosed patients with colon or rectal cancer were recruited in 11 hospitals in the Netherlands. Hospital staff invited eligible patients during a routine clinical visit before the start of treatment. Patients were considered not eligible when they had a history of CRC, a previous (partial) bowel resection, known hereditary CRC, inflammatory bowel disease, dementia or another mental condition limiting their ability to fill out surveys, or were non-Dutch speaking. All study participants provided written informed consent and the study was approved by the Committee on Research involving Human Subjects, region Arnhem-Nijmegen, the Netherlands.

Recurrence data were available for 1605 participants diagnosed between 2010 and 2016. Exclusions were made for the following reasons: missing stage (*n* = 73), stage IV disease (*n* = 132), ASA physical status classification IV (severe systemic disease that is a constant threat to life) (*n* = 3), BMI <18.5 kg/m^2^ (*n* = 8), or CRC recurrence before lifestyle assessment (*n* = 11). Furthermore, we excluded 198 participants who had missing lifestyle data six months after diagnosis. The final sample size for the analyses was 1180.

### 2.2. Lifestyle Assessment

Lifestyle data were collected six months after diagnosis. Habitual dietary intake was assessed with a 204-item semi-quantitative food frequency questionnaire (FFQ). The reference period for the FFQ was the previous month. To assess amounts of food intake, we combined frequencies of intake with standard portion sizes and household measures [33]. The FFQ was previously validated [34], and slightly adapted to be able to distinguish meat intake with respect to red, processed, and white meat. Self-reported dietary intake data from the FFQ were converted into energy, macronutrient, fiber, and alcohol intake based on the 2011 Dutch food composition table [35]. In our RSF models, we included all dietary components (food groups and dietary fiber) present in either the cancer prevention recommendations (American Cancer Society (ACS) [4] or the World Cancer Research Fund/American Institute for Cancer Research (WCRF/AICR) [3]) or national dietary guidelines from the Netherlands [36,37]. Furthermore, we included additional food groups not included in these recommendations (for example, coffee intake) and macronutrients. In total, 44 dietary variables were included in the RSF models (Appendix A).

In addition to the FFQ, participants filled out other lifestyle questionnaires assessing self-reported weight, height, and physical activity, and current smoking (including number of cigarettes smoked per day). Waist (midway between the lowest rib and the iliac crest) and hip circumference were measured with a tape sent to participants. Waist-hip-ratio and BMI (kg/m^2^) were computed. Waist circumference and waist-hip-ratio were standardized to relative values that express excess adiposity directly. Standardizing was done by subtracting the sex-specific cut-offs that determine excess adiposity from the measured values (Appendix A). Moderate-to-vigorous physical activity was self-reported by the validated SQUASH questionnaire [38,39,40]. Moderate-to-vigorous physical activity included all activities (walking, cycling, gardening, odd-jobs, sports, household activities, and work) with a metabolic equivalent value ≥ 3 [41]. To ensure the quality of the data, we checked each questionnaire after completion and contacted participants by telephone for clarification if needed. In total, 6 physical activity, 3 adiposity, and 2 smoking variables were included in the RSF models (Appendix A).

### 2.3. Assessment of Background Variables

Information was obtained on socio-demographic and clinical factors. Socio-demographic information and daily use of nonsteroidal anti-inflammatory drugs (NSAIDs). Clinical data, such as CRC stage, tumor site, administration of neo-adjuvant treatment, adjuvant chemotherapy, and ASA physical status classification were retrieved from the Dutch ColoRectal Audit (Dutch Institute for Clinical Auditing, Leiden, the Netherlands). The Dutch ColoRectal Audit is a nationwide audit initiated by the Association of Surgeons from the Netherlands to monitor, evaluate, and improve CRC care [42]. Self-reported smoking status at diagnosis (never, former, smoking at diagnosis) was also included as a background variable, instead as a lifestyle variable, as smoking status is a potential confounder. In total, 11 background variables were included in the RSF models (see Table 1).

### 2.4. Outcome Assessment

We defined CRC recurrence as time from postdiagnosis lifestyle assessment to locoregional recurrence or distant metastasis. Patients who died without CRC recurrence or who experienced another type of cancer with metastasis were censored in analyses with CRC recurrence as the outcome. Information on recurrences was collected from medical records by trained registrars from the Dutch Cancer Registry from January to March 2018. We defined all-cause mortality as time from postdiagnosis lifestyle assessment to death. Vital status and date of death were determined through linkage to the Municipal Personal Record Database of the Netherlands through December 2019.

### 2.5. Random Survival Forests

Random survival forest (RSF) analysis is an ensemble tree method for the analysis of right censored survival data [25]. Trees in a survival forest are grown randomly using a two-step randomization process (Figure 1). First, each tree is grown using a randomly drawn bootstrap sample (training set), which includes on average two thirds of the original data. Second, random variable selection is used when growing the tree. At each split, a new random subset of candidate variables is selected. The bootstrap sample, including for each tree a random subset of the study population, can be seen as the root of the tree. During the tree-growing process, the root is split into two branches. The branch is split using the variable, from the randomly selected subset of candidate variables, that indicates the largest survival difference between daughter branches. Averaging over trees in combination with the randomization used in growing a tree creates an ensemble of independent trees that form the RSF.

Once an RSF model is computed, prediction accuracy and variable importance can be assessed. Prediction accuracy for RSF was assessed using data that were not included in the tree-growing process (i.e., the remaining one third of the original data) [25]. These data are called out-of-bag (OOB) data (i.e., test set). The RSF prediction error rate has values between 0 and 1, where a lower RSF prediction error rate corresponds to an RSF model with more precise prediction accuracy.

Variable importance (VIMP) was determined by applying the RSF model on the OOB data (i.e., test set) [25]. VIMP is calculated as follows: (i) In the OOB cases for a tree, all values of a certain variable are randomly permuted; (ii) this new variable is put down the tree and a new internal error rate is computed; (iii) the amount this new error rate exceeds the original OOB error is defined as the importance of that variable for the tree; (iv) averaging over the forest yields VIMP. High positive VIMP values indicate that a variable is important in predicting the outcome of interest.

### 2.6. Statistical Analysis

To identify important lifestyle behaviors for recurrence or all-cause mortality, we generated RSF models. The full RSF models were applied on the data of all participants, consisting of all lifestyle and background variables. The important variables regarding either recurrence or all-cause mortality were determined as those with positive VIMP values that exceeded the amplitude of the largest negative value (i.e., the dashed line in Figure 2) [43]. However, as the random process involved in building the trees might influence the VIMP observed, we computed 10 repetitions for each RSF model. For each model repetition, we identified which variables were predictive of the outcome based on the VIMP values. Only those lifestyle variables that were identified in ≥7 out of 10 model repetitions were considered important regarding the outcome and were selected for the final model. The final RSF models contained all 11 pre-defined background variables (see Table 1) plus the subset of identified lifestyle variables to account for possible confounding. Additionally, the analyses were repeated in two subsets of the data based on tumor location (colon or rectum).

Final RSF models were used to derive partial (dependence) plots of selected lifestyle variables. Partial plots represent the effect of each lifestyle behavior on predicted (recurrence-free) survival after accounting for the average effects of all variables in the model and can be used to graphically assess the direction and non-linearity of associations [44]. Therefore, partial plots are adjusted for all variables in the final model, similar to multivariable Cox regression models that include confounders. The *y*-axis of the plot shows the risk of (recurrence-free) survival at different levels of dietary intake (x-axis). Therefore, for example, a value of 0.80 should be interpreted as 80% chance of recurrence-free survival (i.e., no recurrence) or, similarly, 20% chance of recurrence. We chose 3- and 5-years’ time curves to be shown in the partial plot. These time points are clinically relevant and in line with the available follow-up time.

To evaluate prediction accuracy, we computed three additional RSF models next to the full and final models described above. These models contained: (i) Background variables only; (ii) lifestyle variables only; and (iii) only randomly generated noise variables. This last model was used to benchmark the prediction error. This model can be seen as a ‘control’ model and did not include any of the original background or lifestyle variables. The noise variables consist of randomly generated values that follow a normal distribution (mean = 0, SD = 1). Eleven noise variables were included in the benchmark (i.e., ‘control’) models, as the models with only background variables also included 11 variables. For each RSF model, 10 repetitions were generated and used to calculate means and standard errors (SE) of prediction error rates of the respective RSF models.

The analyses were conducted with the statistical software R (version 3.6.1), the R-package RandomForestSRC (version 2.9.3) and SAS (SAS Institute Inc., Cary, NC, USA; version 9.4). In the preliminary analysis, we conducted a grid search to determine model parameters (numbers of trees grown in the forest, number of randomly selected candidate variables, and number of unique cases in terminal branches) with optimal predictive power. The results indicated that the default values were adequate, although ≥1000 trees were needed. Therefore, we generated RSF models with 2000 trees. We used the following default values: (i) Log-rank splitting rule = 10 splits per variable; (ii) number of candidate variables = the square root of the total number of exposure variables; (iii) number of unique cases in terminal branches = 15. We dealt with missing data by using the imputation option within the RandomForestSRC package [25].

## 3. Results

Our cohort consisted of 1180 people diagnosed with CRC. Median age at CRC diagnosis was 66 years and 67% of the tumors were located in the colon (Table 1). Stage III disease (44%) was more common than stage II (30%) and stage I disease (26%). We observed 148 recurrences during 2.6-year (IQR 1.7–3.9) median follow-up. A total of 152 patients died during 4.8-year (IQR 3.7–5.8) median follow-up; 55% of people with a recurrence died during follow-up (*n* = 81).

Figure 2 plots the variable importance (VIMP) of all 66 variables (55 lifestyle and 11 background variables) of the full model. The dashed horizontal line separates the predictive variables from the remaining non-predictive variables. The stage of disease is easily seen to be the most predictive variable for recurrence, while this is age for all-cause mortality. However, some variables were inconsistently identified as predictive variables over the 10 repetitions of the RSF models (Table 2).

For recurrence, sugary drink intake was consistently identified as the most predictive lifestyle variable. Saturated fat intake was identified as the predictive lifestyle variable in 8 out of 10 models. The background variables stage, tumor location, adjuvant chemotherapy, and neo-adjuvant therapy were consistently identified as the top 4 most predictive variables. Separate analyses by tumor location showed that sugary drink intake was identified as important variable among people with colon cancer, but not among people with rectal cancer (Appendix A). Saturated fat intake was identified as important variable in both groups.

For all-cause mortality, 3 lifestyle variables were consistently identified as predictive in all model repetitions: Liquid fat and oil, fruit and vegetable, and animal protein intake. Fruit, polyunsaturated fat, potato, and processed meat intake were identified as predictive lifestyle variables in ≥7 out of 10 models. The background variables age and stage were consistently identified as the top 2 most predictive variables, while ASA-classification was predictive in 7 out of 10 models. Separate analyses by tumor location showed that fruit, liquid fat and oil, and fruit and vegetable intake were only identified as important variables among people with colon cancer (Appendix A). In contrast, animal protein, processed meat, and polyunsaturated fat intake were only identified as important variables among people with rectal cancer.

Final RSF models included the subset of identified lifestyle behaviors (recurrence: sugary drink and saturated fat intake; all-cause mortality: liquid fat and oil, fruit and vegetable, animal protein, fruit, polyunsaturated fat, potato, and processed meat intake) together with the 11 pre-defined background variables that were included as potential confounders. Final RSF models had the smallest mean prediction error rates for both recurrence (0.3376) and all-cause mortality (0.3452) of all constructed models (Table 3). This indicates that adding identified lifestyle variables to an RSF model with background variables reduced prediction error. However, adding all available lifestyle variables to the model worsened prediction error.

Direction and non-linearity between identified lifestyle variables and predicted 3 and 5-year recurrence-free survival (Figure 3) or survival (Figure 4) was assessed visually in partial plots. From the plots in Figure 2 we can see that the association between sugary drink intake and recurrence appears to be approximately linear, with higher intakes being associated with lower recurrence-free survival and thus a higher recurrence risk. From the plots in Figure 3, we can see that the associations between the continuous dietary behaviors and survival appear to be non-linear. For example, a non-linear inverse association was observed for fruit intake with most of the risk reduction observed when increasing intake up to about 100 g/day.

## 4. Discussion

Random survival forests (RSF) identified sugary drink intake as most important lifestyle behavior after colorectal cancer diagnosis related to recurrence in our cohort of 1180 patients with stage I–III CRC. Higher intakes of sugary drinks were associated with increased recurrence risk. For all-cause mortality, fruit and vegetable, liquid fat and oil, and animal protein intake were consistently identified as the most predictive lifestyle variables. These lifestyle variables showed non-linear associations with all-cause mortality. Predictive power improved by adding these identified lifestyle variables to RSF models that only included 11 pre-defined background (socio-demographic and clinical) variables.

This was the first study that identified lifestyle behaviors important for recurrence and all-cause mortality in cancer survivors with a data-driven method. Therefore, we can only compare our results with prospective cohort studies, which assessed associations between post-diagnosis lifestyle behaviors and CRC outcomes with traditional Cox regression models. Our RSF models identified higher sugary drink intake after CRC diagnosis as an important risk factor for recurrence, which is in line with the only previous study which assessed this association among colon cancer survivors [21]. However, sugary drink intake might not be related to recurrence risk among rectal cancer survivors (Appendix A). Further analyses in other cohorts of CRC survivors are needed to support (or refute) the potential role of sugary drink intake in CRC recurrence.

Our RSF model identified three dietary behaviors-fruit and vegetable, liquid fat and oil, and animal protein intake-as important lifestyle behaviors regarding all-cause mortality. These dietary behaviors were selected from a set of 55 lifestyle variables, which included well-known risk factors for cancer incidence which are potentially also linked to CRC survival, as well as lifestyle variables not previously linked to CRC survival. In line with our findings, two previous studies also reported that lower fruit and vegetable intake after CRC diagnosis was associated with higher all-cause mortality [14,16], while no associations were reported for either fruit or vegetable intake in another study [15]. Our partial plots suggest that particularly low fruit and vegetables intake is associated with higher all-cause mortality. Although this is comparable to the inverse non-linear association observed for CRC risk [45], this has not been observed for CRC survival before. Two previous studies in which the association between fat intake and all-cause mortality among CRC survivors was assessed reported mixed findings [46,47]. Although both did not report on liquid fat and oil intake, one concluded that neither total nor major types of dietary fat were associated with disease-free survival [46]. The other study concluded that replacing carbohydrates with plant or polyunsaturated fat was associated with lower all-cause mortality [47]. Previous studies that assessed the association between animal protein intake and all-cause mortality among CRC survivors also reported mixed findings. Replacing carbohydrates with animal protein was associated with a higher all-cause mortality [47]. Instead of animal protein intake, other studies investigated red and processed meat or dairy intake. Red and processed meat intake was not associated with all-cause mortality [15,22], while another study reported higher all-cause mortality with lower red and processed meat intake [14]. Higher all-cause mortality was also reported for lower milk intake [48]. A low animal protein intake could result in loss of muscle mass, which could worsen clinical outcomes and increase mortality risk [49,50]. Taken together, emerging evidence seems to indicate low fruit and vegetable intake is associated with higher all-cause mortality, especially among colon cancer survivors. Further research is needed to assess the potentially non-linear associations between liquid fat and oils or animal protein intake and all-cause mortality. Such studies should also assess if these associations differ by tumor location, as our additional analyses identified different lifestyle behaviors among subgroups with colon or rectal cancer.

RSFs are better suited than traditional Cox regression models to identify a subset of exposures that are related to the outcome of interest from a large set of potentially interesting exposures. Researchers can use RSF to consider many lifestyle behaviors simultaneously and to identify which of these modifiable behaviors are most important for CRC recurrence and all-cause mortality. Applying many Cox regression models to test all these associations with either recurrence or all-cause mortality would result in multiple testing. There are two advantages for using RSFs in this situation. First, RSFs do not rely on *p*-values and, more importantly, RSF uses a subset of data not included in model building (i.e., out-of-bag data) to identify important variables. Second, RSF takes complex interactions between variables into account. Cox regression models are a suitable method to test hypothesis on exposure-outcome associations with a limited number of exposures of interest. Cox models are, therefore, complementary to RSF models. Future research, in external cohorts, could use Cox regression models to further study the associations between our identified dietary behaviors and CRC progression.

Several studies have now compared RSF to other methods, including Cox regression models, and these have shown that the predictive accuracy of RSF was consistently better than, or at least as good as, competing methods [25,26,27,28,51]. In our study, predictive accuracy was best in models that included identified lifestyle behaviors in addition to background variables, but performance was only slightly better than our models with only background variables. A similar pattern was observed in a previous study which identified modifiable lifestyle behaviors related to CRC risk in the EPIC-cohort [52]. Their final model, including both age and identified modifiable lifestyle behaviors, also performed only slightly better than a model with age only. However, they showed that lifestyle information in addition to age was important for absolute risk assessment. The reported prediction error rates of our final models are similar to those reported in the EPIC-cohort [52] and several other RSF models [27,51]. However, models which included all 55 lifestyle behaviors performed worse than models with only background variables. We assume that many of these lifestyle behaviors are not impacting CRC prognosis, and thus, add ‘noise’ to the model, which decreases predictive accuracy.

Potential limitations of our study should be considered. A first limitation of our study is that we have not validated our RSF models with an external cohort of CRC survivors. Although RSF does validate the model by testing prediction on the “out-of-bag” sample, that is, individuals that were not used to create the particular tree, the ensemble of trees are still derived from the entire original dataset. This study needs to be repeated in an external cohort to see if the same variables will be identified. This is not different from studies which use Cox regression models, as multiple studies are always needed to strengthen the evidence. Another limitation is that we noted some variations in the identified predictive variables over the 10 repetitions of the RSF models. This variation is likely explained by the conservative approach used to identify important lifestyle variables. All variables below the threshold are clearly not important, while values above the threshold may (or may not) be predictive [43]. To limit this variation, we created larger RSFs with 2000 trees and limited our identified lifestyle variables to those consistently identified in 10 out of 10 model repetitions. Although variable importance values differed slightly between repetitions, our partial plots were robust as we observed no clear differences between partial plots based on slightly different models. Furthermore, we could not explore cause-specific mortality, as we do not have access to these data. This would have been interesting, as we identified three dietary behaviors (fruit and vegetable, liquid fat and oil, and animal protein intake) related to all-cause mortality, which were not important for CRC recurrence. These dietary behaviors might, thus, specifically be related to other causes of death (e.g., cardiovascular mortality or mortality associated with loss of muscle mass), but not with CRC-specific mortality. Last, we did not include information on muscle mass, as this information is only available at diagnosis for a subset of our population. Our group previously showed that muscle mass tended to increase with increasing BMI among stage I-III CRC patients [53]. Thus, lower BMIs might serve as a proxy for low muscle mass in the current analyses. However, BMI was not identified as an important variable for all-cause mortality.

Strengths of the current study include the availability of both CRC recurrence data and a large number of post-diagnosis lifestyle behaviors related to diet, physical activity, alcohol use, adiposity, and smoking, which allowed us to simultaneously identify which of these behaviors are related to CRC outcomes. This was the first study that considered many modifiable lifestyle behaviors simultaneously to identify modifiable risk factor for CRC recurrence and survival. The results of this study indicate the relative importance of different lifestyle behaviors and show that lifestyle behaviors currently not included in the recommendations could also impact CRC prognosis.

## 5. Conclusions

This study among CRC patients with non-metastatic disease identified different lifestyle behaviors for recurrence risk and all-cause mortality. For recurrence, higher intakes of sugary drinks were associated with increased recurrence risk. For all-cause mortality, fruit and vegetable, liquid fat and oil, and animal protein intake were identified as most important lifestyle behaviors. These latter behaviors showed non-linear associations with all-cause mortality. Identified behaviors comprised a few known factors, included in cancer prevention recommendations, but also some additional lifestyle behaviors. Our exploratory RSF findings give new ideas on the potential associations between certain lifestyle behaviors and CRC prognosis that still need to be confirmed in other cohorts of CRC survivors.

## Figures and Tables

**Figure 1 cancers-13-02442-f001:**
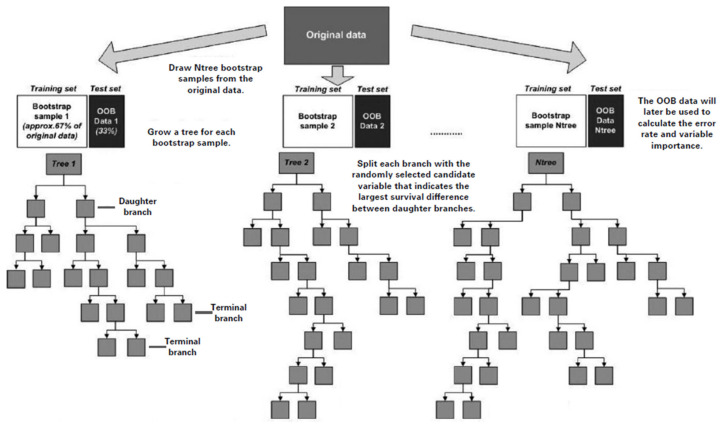
Graphical presentation of the Random Survival Forest (RSF) algorithm. Adapted from Datema et al., 2012 [26]. OOB, out-of-bag data.

**Figure 2 cancers-13-02442-f002:**
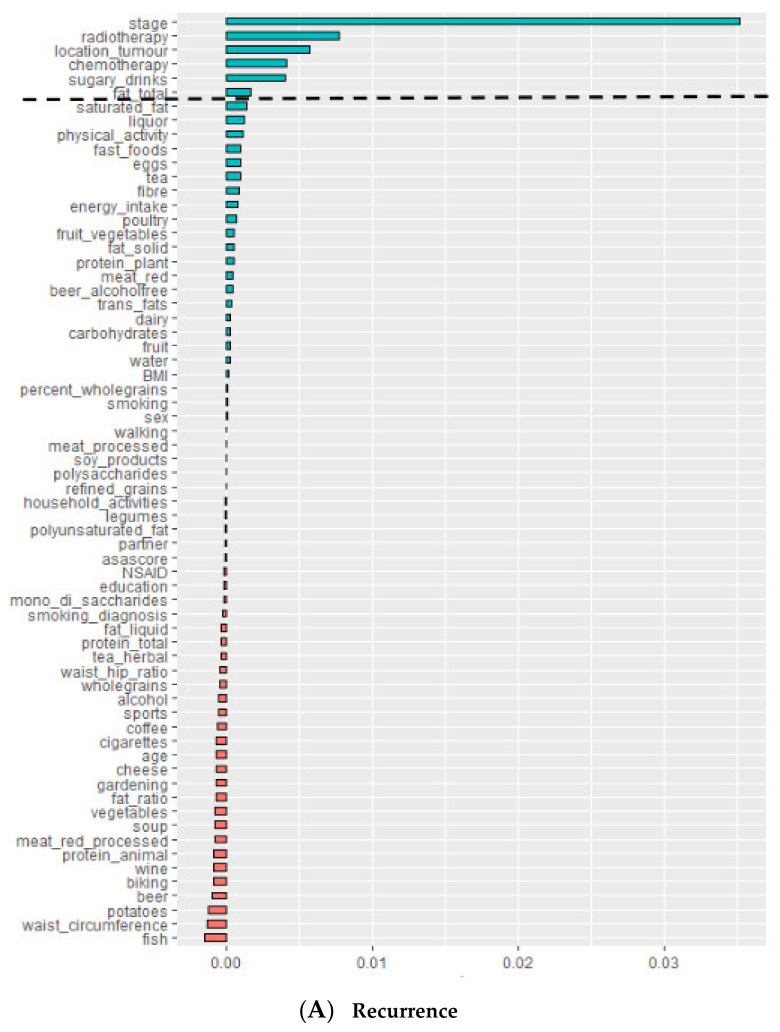
Variable importance (VIMP) from random survival forest analysis for (**A**) colorectal cancer recurrence, and (**B**) all-cause mortality for one model repetition. The dashed horizontal line is the threshold for filtering variables: All variable above the line are identified as predictive variables. VIMP values are shown for 1 out of 10 model repetitions. Notably, some variations in VIMP values were noted over the 10 repetitions of the RSF models.

**Figure 3 cancers-13-02442-f003:**
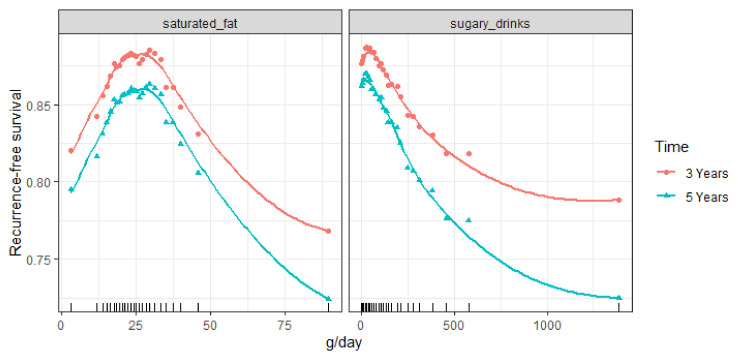
Partial plots of identified lifestyle variables for recurrence. Values on the vertical axis represent predicted three-year and five-year recurrence-free survival for a given variable after adjusting for all other variables (background and shown lifestyle variables). Dietary intakes in grams per day are on the horizontal axis. A lower predicted recurrence-free survival means a higher risk to develop a local or distant recurrence within three or five years of follow-up. The rug plots on the x-axis show the distribution of intake data observed in the cohort; about 90% of observations occurs between the second and second-last rug.

**Figure 4 cancers-13-02442-f004:**
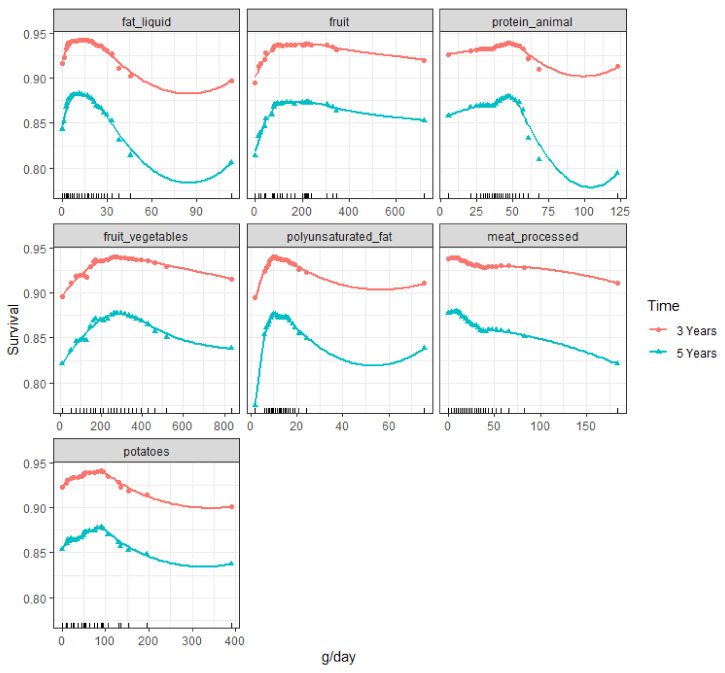
Partial plots of identified lifestyle variables for all-cause mortality. Values on the vertical axis represent predicted three-year and five-year survival for a given variable after adjusting for all other variables (background and shown lifestyle variables). Dietary intakes in grams per day are on the horizontal axis. The rug plots on the x-axis show the distribution of intake data observed in the cohort; about 90% of observations occurs between the second and second-last rug.

**Table 1 cancers-13-02442-t001:** Characteristics of the study population at colorectal cancer diagnosis and lifestyle characteristics six months after diagnosis.

Background Variables, *n*(%) or Median (IQR)	*n* = 1180
Age at diagnosis, y	66 (61–71)
Men	747 (63%)
Education (missing *n* = 9)	
Low	482 (41%)
Medium	314 (27%)
High	375 (32%)
Living with partner (missing *n* = 7)	988 (84%)
Tumor stage	
I	307 (26%)
II	356 (30%)
III	517 (44%)
Tumor site	
Colon	796 (67%)
Rectum	384 (33%)
Neo-adjuvant treatment	272 (23%)
Adjuvant chemotherapy	284 (24%)
ASA physical performance classification (missing *n* = 51)	
I	354 (30%)
II	653 (55%)
III	122 (10%)
Daily NSAID use	102 (9%)
Smoking at diagnosis (missing *n* = 8)	
Yes	119 (10%)
Former	694 (59%)
Never	359 (31%)
**Lifestyle six months post-diagnosis, *n*(%) or median (IQR)**	
Body Mass Index, kg/m^2^ (missing *n* = 6)	25.9 (23.9–28.5)
Physical activity ^1^, min/wk.	480 (240–840)
Diet	
Fruits and vegetables, g/day	248 (147–350)
Red and processed meat, g/day	63 (38–85)
Sugary drinks, g/day	70 (13–176)
Dietary fiber, g/day	19 (15–24)
Energy intake, kcal/day	1765 (1472–2112)
Alcohol intake	
Non-drinker ^2^	293 (25%)
Amount (g/d) among drinkers	9 (3–21)
Amount (g/d) among all	5 (0–16)
Current smoker (missing *n* = 2)	80 (7%)

^1^ Moderate-to-vigorous physical activity included all activities with a metabolic equivalent value ≥ 3; ^2^ No alcohol intake in past month. Bold: sub-heading.

**Table 2 cancers-13-02442-t002:** Variables predictive of recurrence or all-cause mortality based on variable importance.

Variables Predictive of Recurrence	Number of Times Selected as Predictive Variable in 10 Repetitions of RSF Model
*Stage*	10
*Tumor location*	10
*Adjuvant chemotherapy*	10
*Neo-adjuvant therapy*	10
Sugary drinks	10
Saturated fat	8
Fruit	6
Total fat	4
Trans-fats	3
Eggs	3
Polyunsaturated fat	3
Carbohydrates	3
Fiber	2
Liquor	2
Energy intake	2
**Variables predictive of all-cause mortality**	
*Age*	10
*Stage*	10
Liquid fat & oils	10
Fruit & vegetables	10
Animal protein	10
Fruit	9
Polyunsaturated fat	9
Potato	8
Processed meat	8
*ASA classification*	7
Herbal tea	6
Sugary drinks	6
Soup	6
*Adjuvant chemotherapy*	6
Alcohol	5
BMI	4
Beer	4
*Education*	4
Plant protein	4
*Neo-adjuvant therapy*	2
Dietary fiber	2

Variables printed in *italics* are background variables, all other variables are lifestyle variables. Variables were selected as predictive based on their VIMP values. Only variables selected in ≥2 model repetitions are included in this table. Bold: sub-heading.

**Table 3 cancers-13-02442-t003:** RSF-derived error rates for the prediction of recurrence and all-cause mortality in different RSF models based on 10 model repetitions.

RSF Model	Prediction Error Rate ^1^ (Mean ± SE)	
	Recurrence	All-Cause Mortality
Final model (background and identified lifestyle variables)	0.3376 ± 0.0005	0.3452 ± 0.0006
Only background variables	0.3570 ± 0.0005	0.3483 ± 0.0004
Full model (background and lifestyle variables)	0.3777 ± 0.0006	0.3964 ± 0.0009
Only lifestyle variables	0.4858 ± 0.0014	0.4309 ± 0.0007
Only noise (benchmark model)	0.5706 ± 0.0014	0.4886 ± 0.0011

Background variables included age, sex, education, living with partner, stage of disease, neo-adjuvant treatment, adjuvant chemotherapy, tumor location, smoking status at diagnosis, use of non-steroidal anti-inflammatory drugs at diagnosis, and ASA classification. ^1^ Standard error (SE) represents randomness based on 10 repetitions of the RSF model within the same dataset.

## Data Availability

The data presented in this study are available on request from the corresponding author. The data are not publicly available because the data consist of identifying cohort information.

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
