# Peer review of "Identification of Lifestyle Behaviors Associated with Recurrence and Survival in Colorectal Cancer Patients Using Random Survival Forests"

_cancers, 2021, doi:10.3390/cancers13102442_

Round 1
Reviewer 1 Report
This paper illustrates the use of a somewhat novel, not-yet-validated statistical model, random survival forests (RSF), on a specific cancer population in order to identify certain lifestyle behaviors that might affect its survival outcome. The chosen population consists of newly diagnosed stage I-III CRC patients from a prospective cohort study. A number of lifestyle behaviors including various diets and their particulars, physical activity, obesity, alcohol and smoking were evaluated for their predictive risks in cancer recurrence and all-cause (not necessarily related to cancer) mortality.
It identifies that high intake of sugary drinks is associated with increased cancer recurrence risk. For all-cause mortality, it identifies intake of fruit and vegetable, liquid fat & oil, and animal protein as significantly predictors in these patients.
The paper is well written and the statistical method (RSF) is described in great details and proposed as an alternative to the often used multivariable Cox proportional hazard regression models. The authors indicated that their RSF model has not been validated using an external cohort of CRC survivors. If it were validated, the study would have offered stronger conclusions regarding these same lifestyle variables.
Overall, it is a commendable article if a biostatistician (1) could further examine the strength of its novel statistical method RSF, and (2) could determine whether or not a small external validation cohort is needed.
Reviewer 2 Report
Thank you for giving me the opportunity to review the paper by van Zutphen et al - Identification of lifestyle behaviors associated with recurrence and survival in colorectal cancer patients using random
4 survival forests. The major PRO and novelty of this study is that the authors have decided to use RSF as an alternative to the more common Cox multivariate regression models (more rigid and sometimes with statistical biases). This is a fair study, however we feel that it still can be improved.
MAJOR comments:
- The ”Introduction” Subchapter needs extensive, well-documented and balanced data concerning risk and protective factors related to the development and progression of CRC.
- It is very uncommon and somehow unclear why the authors have decided to present first-handedly the ”Results” section of the paper a priori of the ”Materials/Patients and Methods” section. The ”IMRAD” structure of a scientific medical paper needs to be respected, from our point of view.
- It is to our best belief that colon and rectal cancer need to be addressed as separated pathological entities, since their risk factors, epigenetic mechanisms, natural history, multidisciplinary management and outcome differ. Furthermore, from our perspective, this issue needs to be noted in the revision format of the paper.
- The “Discussion” section of the paper did not address the “hot topic” of pre/post-treatment patient frailty. Abundant literature is available on this. Discussing frailty could add strength to the results and conclusions on the paper. Frailty and muscle mass are significant factors when predicting 30-day morbidity and mortality rates, 3, 5-yrs Survival rates and OS survival rates.
MINOR comments:
- Lines 27-28 need rephrasing – redundancy and repetitions
- Line 30 “&” can be avoided
- Lines 88-92 – please clearly underline the aim of the study
Reviewer 3 Report
In the current study, the authors aimed to evaluate the association between lifestyle behaviours and both cancer recurrence and overall survival in colorectal cancer patients. The authors used random forests in order to take into account several potential predictors of outcomes, taking into account background (baseline) covariates. The methodology is appropriate and the manuscript is very well written. However, I would ask the authors:
- Lifestyle data were collected six months after cancer diagnosis, and they referred to the previous months. My major concern is that cancer patients may substantially change their lifestyle behaviour after a cancer diagnosis (i.e. they may stop drinking alcohol and eating unhealthy food after diagnosis). However, I expect that behavioural changes happened just after cancer diagnosis have only a small impact on cancer recurrence and overall survival, while pre-diagnosis behaviours may have a larger impact. For instance, let us compare two patients, both non-drinkers after cancer diagnosis, but one non-drinker also before diagnosis and the other one heavy drinker before cancer diagnosis. I expect the first patients to have a lower risk of death, as compared to the second one. In other terms, my concern is that assessing lifestyle behaviours at baseline (i.e. referred to usual behaviour before cancer diagnosis) may better predict the outcomes than lifestyle behaviour assessed after cancer diagnosis.
- Figure 4 is not cited in the text. Also, please check Figure 4 title.
Round 2
Reviewer 1 Report
The second version is much improved, especially with the extensive revision of the Method section.
Reviewer 3 Report
The authors replied satisfying my questions. The paper is now suitable for publication.